# Comparison of Knee Muscular Strength Balance among Pre- and Post-Puberty Adolescent Swimmers: A Cross-Sectional Pilot Study

**DOI:** 10.3390/healthcare11050744

**Published:** 2023-03-03

**Authors:** Bruno Lombardi Amado, Claudio Andre Barbosa De Lira, Rodrigo Luiz Vancini, Pedro Forte, Taline Costa, Katja Weiss, Beat Knechtle, Marilia Santos Andrade

**Affiliations:** 1Sports Medicine Residency Program, Department of Orthopedics and Traumatology, Federal University of São Paulo, São Paulo 04021-001, Brazil; 2Human and Exercise Physiology Division, Faculty of Physical Education and Dance, Federal University of Goiás, Goiânia 74690-900, Brazil; 3Center for Physical Education and Sports, Federal University of Espírito Santo, Vitória 29075-910, Brazil; 4Department of Sport, Higher Institute of Educational Sciences of the Douro, 4560-547 Porto, Portugal; 5Department of Sports, Instituto Politécnico de Bragança, 5300-253 Bragança, Portugal; 6Research Center in Sports, Health and Human Development, 7000-671 Covilhã, Portugal; 7Department of Physiology, Federal University of São Paulo, São Paulo 04021-001, Brazil; 8Institute of Primary Care, University of Zurich, 8091 Zürich, Switzerland

**Keywords:** puberty, isokinetic, peak torque, conventional ratio, muscle balance

## Abstract

Muscular weakness and strength imbalance between the thigh muscles are considered risk factors for knee injuries. Hormonal changes, characteristic of puberty, strongly affect muscle strength; however, it is unknown whether they affect muscular strength balance. The present study aimed to compare knee flexor strength, knee extensor strength, and strength balance ratio, called the conventional ratio (CR), between prepubertal and postpubertal swimmers of both sexes. A total of 56 boys and 22 girls aged between 10 and 20 years participated in the study. Peak torque, CR, and body composition were measured using an isokinetic dynamometer and dual-energy X-ray absorptiometry, respectively. The postpubertal boys group presented significantly higher fat-free mass (*p* < 0.001) and lower fat mass (*p* = 0.001) than the prepubertal group. There were no significant differences among the female swimmers. Peak torque for both flexor and extensor muscles was significantly greater in postpubertal male (*p* < 0.001, both) and female swimmers (*p* < 0.001 and *p* = 0.001, respectively) than in prepubertal swimmers. The CR did not differ between the pre- and postpubertal groups. However, the mean CR values were lower than the literature recommendations, which brings attention to a higher risk of knee injuries.

## 1. Introduction

Since the beginning of the 20th century, strength training for swimmers has been discussed among coaches and researchers, but there is evidence of the importance of propulsion in water and swimming performance [1]. Several previous studies have shown a positive correlation between muscular strength and swimming performance in short-distance (shorter than 400 m) [2] and long-distance events (longer than 400 m) [3,4]. 

Despite the undoubted importance of upper limb strength for performance during the free-swimming phase, lower limb strength is of fundamental importance at the start and the turns. Indeed, a significant correlation between lower-limb strength and turn times [5,6,7] or time to reach 5 or 10 m after the start [7,8] has already been demonstrated.

In addition, there are strength training recommendations for swimmers to improve bone mineral density (BMD), especially among children and adolescent athletes [9,10], because swimmers have been demonstrated to have lower BMD than other athletes who perform weight-bearing exercises, such as rhythmic gymnasts [11] or footballers [12]. 

Strength training has also been recommended as a part of an injury prevention program [13]. Tooth et al. [14] conducted a review of risk factors for sports injuries and demonstrated that muscular weakness is a very important risk factor for injuries. Muscular strength balance plays an important role in joint stability [1,15]. Thus, the isokinetic strength profile and strength balance ratios between antagonistic muscles have been described for several sports and joints [16,17,18,19]. Therefore, lower limb strength plays a significant role, as the knees are one of the most affected joints in swimmers, in addition to the shoulder and lumbar spine [13].

For the knee joint, the strength balance ratio between the flexor and extensor muscles has been measured to assess joint stability [15,20,21,22]. Traditionally, the strength balance has been calculated from the isokinetic peak torque value of knee flexor muscles divided by the isokinetic peak torque value of knee extensor muscles, both in concentric action (flexor-to-extensor ratio), and this ratio has been called the conventional balance ratio (CR) [23]. Values for CR higher than 60% have been generally accepted as preventive measures for injuries [23]. However, athletes from different sports commonly present some degree of strength imbalance because of their unique and repetitive muscular demands or lateral preferences [22].

Another factor that could affect strength balance among adolescent athletes is the pubertal phase. From puberty, a marked gain in muscle strength is evident [24,25]. However, it is unknown whether the abrupt muscular strengthening that occurs during puberty is a balanced gain in all the muscles of the lower limbs or whether the specific and repetitive muscular demand (numerous turns during swimming training) added to the abrupt strength gain characteristic of puberty can generate an imbalance between knee flexor and extensor muscles.

Therefore, the aim of the present study was to compare knee muscle strength, the ratio between the knee flexor and extensor muscles, and body composition between prepubertal and postpubertal swimmers of both sexes. It was hypothesized that the prepubertal participants would present less imbalance in comparison to the postpubertal group.

## 2. Materials and Methods

### 2.1. Design and Procedure

This was a cross-sectional study. Participants attended the laboratory once and were asked to refrain from strenuous workouts on the day before the visit. Sexual maturity was evaluated using the Tanner scale [24], and an isokinetic strength test for thigh muscles and body composition tests were performed. All the tests were performed in the morning between 9 a.m. and 11 a.m. Athletes and guardians were informed of the potential risks and benefits of the study and signed an informed consent form to participate in this study. All experimental procedures were approved by the University Human Research Ethics Committee (no.0503/2017) and conformed to the principles of the Declaration of Helsinki.

### 2.2. Participants

Seventy-eight swimmers (56 boys and 22 girls) participated in this study. All the athletes from the Olympic Center of Training and Research swimming team were recruited to participate in the study. The volunteers were divided into two groups according to sexual maturity and sex (pre- and postpuberty). Group 1 consisted of athletes presenting Tanner scale G1, G2, or G3 for boys and M1, M2, or M3 for girls. Group 2 consisted of athletes with Tanner scale G4 or G5 for boys and M4 or M5 for girls. The cut-off for separating the groups was supported by the fact that Tanner scale 4 is the point of puberty development that causes a significant increase in strength [25]. The inclusion criteria were as follows: participation in swimming training for at least 1 year, five times a week, and at least 1 h per day. The exclusion criteria were as follows: taking any medication, presenting pain in the lower limbs in the last 6 months, having previous lower limb surgery, or presenting any disease that could impact muscular strength. No athletes were excluded.

### 2.3. Evaluation of Sexual Maturity

The sexual maturity of the athletes was evaluated by a pediatrician based on the criteria defined by Tanner [24]. These criteria include pubic hair development (P1, P2, P3, P4, and P5) in both sexes and the quantity and distribution of secondary sexual characteristics according to the developmental stage of the genitals in boys (G1, G2, G3, G4, and G5) and breast development in girls (M1, M2, M3, M4, and M5). Stage 1 (P1, G1, and M1) corresponds to the prepubertal stage, and stage 5 (P5, G5, and M5) corresponds to the complete development stage.

### 2.4. Body Composition

The body composition of the participants was evaluated using dual-energy X-ray absorptiometry (with the software 12.3, Lunar DPX, Madison, WI, USA), which has already been shown to be a reliable method [26] for obtaining data about fat-free mass (FFM) mass and fat mass (FM). These variables were expressed in kilograms (kg) and percentages (%). The values were obtained with the participants in a supine position, centrally aligned with 10 cm between the feet and 5 cm between the hands and trunk. Participants were instructed to wear comfortable clothes without metal objects such as snaps, belts, jewelry, zippers, etc. and to eat and drink as they normally would. They were all evaluated after bladder voiding. The same experienced examiner performed all tests. 

### 2.5. Isokinetic Strength Test

Before the isokinetic strength test, all participants performed a 5-min warm-up on a cycle ergometer (Cybex Inc., Ronkonkoma, NY, USA) at a resistance of 25 W, followed by low-intensity dynamic stretching exercises to avoid muscular strength decrease [27].

After the warm-up, an isokinetic strength test for the knee flexor and extensor muscles of the dominant lower limb was performed using an isokinetic dynamometer (Biodex Medical Systems Inc., Shirley, NY, USA). Lower limb dominance was determined by asking participants which limb they preferred to use when kicking a ball. The test position was determined according to the manufacturer’s instructions. The participants remained in the sitting position with their hips flexed at approximately 85°, and standard stabilization straps were placed around the trunk, waist, and distal femur of the dominant limb to minimize additional movement, secure the test, and perform all the tests under the same conditions. The axis of the dynamometer was visually aligned with the lateral femoral condyle while the knees were flexed at 90°. The resistance pad was placed as distally as possible on the lower limb, and the knee was tested from 5° to 95° of knee flexion (full knee extension defined as 0°). Gravity correction was performed according to the manufacturer’s specifications. 

The participants performed five sequenced maximal repetitions of knee flexion and extension at 60°/s to measure the peak torque (PT) (Nm). The conventional ratio (CR) ratio was calculated from the values of the peak torque of knee flexor muscles/peak torque of knee extensor muscles, both in concentric action at 60 deg/sec. Data were stored for analysis. An angular speed of 60°/s was selected as the lowest speed to avoid high joint pressure while producing the highest torque. The same verbal encouragement was provided to each participant throughout the test [28].

### 2.6. Statistics

The analyses were performed using Jamovi (Jamovi project, version 2.2.5 [Computer Software]). All variables presented normal distribution and homogeneous variability according to the Shapiro–Wilk and Levene tests, respectively. Data are presented as the mean and standard deviation. To compare anthropometric and isokinetic strength data between the two groups, Student’s *t*-test for independent samples was used. The measures of Cohen’s effect size (d) for differences between the groups were determined by calculating the mean difference between the two sexes and then dividing the result by the pooled standard deviation. Calculating effect sizes, the magnitude of any difference was judged according to the following criteria: d = 0.2–0.4 was considered a “small” effect size; 0.5 to 0.7 represented a “medium” effect size; and higher than 0.8 a “large” effect size [29]. The G*Power version 3.1.9.2 (a program written, conceptualized, and designed by Franz, Universitat Kiel, Germany; freely available windows application software) was used for the power analysis calculation [30]. For the power analysis, the *t*-test family was selected, and the mean and standard deviation values for all measured variables for both groups, as well as the effect size values (Cohen d), were included for power analysis. Statistical power ranged from 0 to 1, with higher values indicating a higher chance of correctly rejecting the null hypothesis when a specific alternative hypothesis is true. The basis desired power is usually set to 0.80 as a convention [29]. The level of significance was set at *p* < 0.05.

## 3. Results

As expected, the age (years), body mass (kg), and height (cm) of the prepubertal boys group were significantly lower than the values for the postpubertal group (Table 1).

The girls also presented values for age, body mass, and height significantly higher for postpuberal than for prepubertal groups (Table 2).

The FFM (kg) was significantly higher, and FM (%) was significantly lower in the boys postpubertal group than in the prepubertal group (Table 3). 

Conversely, puberty presented a different effect in the girls group. The FFM (kg) and FM (%) were not different between the prepubertal and postpubertal groups (Table 4).

In the boys groups, the PT values for the knee extensor muscles were significantly higher in the postpubertal group than in the prepubertal group. In the same way, the PT for the knee flexor muscles also was higher in the postpubertal group than in the prepubertal group. In addition, the effect size of the PT differences was similar for the knee extensor and flexor muscles. Considering the CR between the flexor and extensor PT values, there was no significant difference between the boys groups (Table 3).

In the girls groups, the PT for the knee extensor muscles was also higher in the postpubertal group than in the prepubertal group. A similar pattern was observed in the knee flexor muscles, where the postpubertal group presented significantly higher values than the prepubertal group. However, it could be seen that the effect size of the knee flexor muscles difference was higher. Consequently, the CR between the flexor and extensor PT values tended to be higher in the postpubertal group than in the prepubertal group, although the difference did not reach the significance threshold (Table 4).

## 4. Discussion

The aim of this study was to evaluate and compare knee muscle strength and CR between thigh muscles between prepubertal and postpubertal swimmers. The main results of the present study were as follows: (i) postpubertal male swimmers presented higher FFM (kg) and lower FM (%) than prepubertal swimmers; (ii) FFM (kg) and FM (%) were not different between prepubertal and postpubertal girls groups; (iii) postpubertal swimmers from both sexes showed greater knee flexor and extensor PT values than prepubertal swimmers; (iv) prepubertal and postpubertal swimmers, from both sexes, were not different in CR. Therefore, the initial hypothesis of the present study was not confirmed as the CR was not affected by puberty, neither in male nor female athletes.

FFM is a variable that is affected by puberty, especially in boys. Male puberty is characterized by increased levels of circulating testosterone, which is the hormone responsible for promoting sexual function development and male somatic characteristics, including higher muscle mass [31,32]. In contrast to boys, puberty in girls is affected by the sex hormone estrogen, which triggers the release of gonadotropin-releasing hormone and the subsequent activation of the hypothalamic-pituitary-gonadal axis [31]. A higher estrogen level is associated with an increase in FM but has no effect on FFM. This justifies the fact that the boys presented an expressive increase in FFM after puberty, whereas the girls did not, as previously demonstrated by Fukunaga et al. [33] and Costa et al. [25]. 

Considering FM (%) in the boys groups, a significant decrease was observed, which can be mathematically explained by the significant increase in absolute values of FFM (kg); therefore, the percentage of FM decreased. In addition, the literature supports these data because of the intense activity of the hypothalamic-pituitary-gonadal axis during this period of life [31]. Schneider et al. also reported this change in body composition, with the FM percentage varying from 21.0% to 13.6% in male swimmers after puberty, which is in congruence with the present data (14.3% FM in the postpubertal boys group) [34]. 

FM (%) did not differ between the girls groups. For female nonathletes, an increase in FM (%) after puberty is expected because of the higher estrogen levels, which directly influence fat accumulation [35]. However, the participants in the present study were athletes, and despite the higher estrogen levels, the participants were used to training at least 5 days a week, 1 h per day, which certainly helps to maintain FM (%) at lower levels, despite the present results (26.1% for FM in the postpubertal girls group) being higher than the values reported by Schneider et al., which were 20.7–18.7% in the postpubertal female swimmers [34].

According to the strength values, the present results showed significantly higher values for knee flexor and extensor muscle PT in the postpubertal groups than in the prepubertal boys group. There were significant differences between the groups in PT, with the postpubertal groups having higher values in both extension and flexion at 60°/s. These data are consistent with the acknowledgment of the strength increase that occurs during puberty, partly due to the elevation in the serum levels of anabolic hormones, such as growth hormone and testosterone [36,37]. Of course, these hormones play an important role in the development of neuromuscular maturation, but the improvement of the central nervous system during the child’s development (especially in those who have more physical abilities) is another variable to be considered [25]. These findings corroborate previous studies that compared PT between prepubertal and postpubertal adolescents [33,34]. It is important to note that the effect size of puberty on the difference in the strength of the extensor and flexor muscles is quite similar, indicating that both muscle groups became stronger in a similar way. Consequently, no difference was observed in the CR between the flexor and extensor muscles. Nevertheless, the mean CR values presented by these swimmers deserve further attention. Although the optimum CR between the flexor and extensor muscles has not been reported, values approaching 60% (measured at 60°/sec during concentric action) are recommended for injury prevention during dynamic movements [23,38]. Values below 60% were previously associated with a higher risk of knee injuries (such as muscular injuries, especially in the hamstrings, anterior cruciate ligament injury, and tendon injuries) [39,40,41]. Thus, the mean CR values for the prepubertal (51.9 ± 7.3%) and postpubertal (53.9 ± 6.7%) groups can be described as relatively low. 

Similar to the boys groups, the CR between the flexor and extensor muscles was not different between the girls groups. However, the situation requires much more attention because prepubertal, and postpubertal mean values for CR were even lower than 60% (46.4 ± 4.8% and 49.5 ± 9.4%, respectively). Similar results were presented by Secchi et al. (2011) for elite swimmers and Dalamitros et al. (2015) for adolescent swimmers, with CR values ranging from 46.5% to 53.4% [42]. 

This study had some limitations. First, the reduced sample size may have contributed to the lack of significant difference between CR for the male groups since the power of statistical analysis was lower than 0.80 (power = 0.70). Conversely, for the female groups, the power for all the analyses, including for CR, were higher than 0.80, which ensures sufficient security to draw conclusions resulting from the results of statistical analyzes. Therefore, the authors recommend that other studies should be done with a larger sample size, mainly for the male group. Second, it was a cross-sectional study, and for a better understanding of how puberty affects muscular strength and balance ratios, a longitudinal study should be performed. Third, sexual maturity was measured using Tanner levels, and more accurate and precise methods should be used to precisely measure this variable.

In conclusion, the flexor and extensor muscles were significantly stronger in postpubertal female and male swimmers than in prepubertal swimmers of the same sex, and the strengthening was balanced between agonist and antagonist muscles once the CR did not change between the groups of the same sex. However, the CR for the female swimmers was lower than the literature recommendation; therefore, this situation deserves attention. The authors of the present study recommend that coaches focus on strengthening the hamstring muscles of female adolescent swimmers to maintain a CR higher than 60%, which is recommended to minimize the risk of knee injury. 

## Figures and Tables

**Table 1 healthcare-11-00744-t001:** Anthropometric data of the boys in the study.

	Pre Puberty (n = 22)	Post Puberty (n = 34)	*p*-Value	Power	Effect Size (d)	CI for Effect Size
Age (years)	11.3 ± 1.2	15.3 ± 2.8	<0.001	>0.99	1.676	0.9 to 2.4
Body mass (kg)	39.7 ± 8.8	61.5 ± 11.4	<0.001	>0.99	2.086	1.2 to 3.0
Height (cm)	149.9 ± 9.91	169.7 ± 9.5	<0.001	>0.99	2.066	1.2 to 3.0

Data are expressed as mean ± standard deviation. CI: confidence interval.

**Table 2 healthcare-11-00744-t002:** Anthropometric data of the girls in the study.

	Pre Puberty (n = 11)	Post Puberty (n = 11)	*p*-Value	Power	Effect Size (d)	CI for Effect Size
Age (years)	11.3 ± 1.2	15.0 ± 1.8	<0.001	>0.99	2.401	1.3 to 3.5
Body mass (kg)	42.4 ± 7.4	54.8 ± 7.4	<0.001	>0.99	1.673	0.9 to 2.4
Height (cm)	153.6 ± 7.4	161.1 ± 4.9	0.012	0.97	1.182	0.4 to 1.8

Data are expressed as mean ± standard deviation. CI: confidence interval.

**Table 3 healthcare-11-00744-t003:** Body composition and strength values for the boys’ groups.

	Pre Puberty (n = 22)	Post Puberty (n = 34)	*p*-Value	Power	Effect Size (d)	CI for Effect Size
Fat-free mass (kg)	29.8 ± 4.8	50.6 ± 9.8	<0.001	>0.99	2.540	1.4 to 3.6
Fat mass (%)	20.4 ± 8.7	14.3 ± 4.6	0.001	0.58	0.941	0.5 to 1.3
PT Ext 60°/s (N·m)	87.2 ± 21.4	155.8 ± 40.9	<0.001	>0.99	1.977	1.1 to 2.9
PT Flex 60°/s (N·m)	45.3 ± 12.3	84.2 ± 24.0	<0.001	>0.99	1.920	1.1 to 2.8
CR (Flex/Ext) (%)	51.9 ± 7.3	53.9 ± 6.7	0.292	0.70	0.291	−0.2 to 0.7

Data are expressed as mean ± standard deviation. CI: confidence interval. PT: peak torque; Ext: extensors; Flex: flexors; CR: conventional ratio.

**Table 4 healthcare-11-00744-t004:** Body composition and strength values for the girls’ groups.

	Pre Puberty (n = 22)	Post Puberty (n = 34)	*p*-Value	Power	Effect Size (d)	CI for Effect Size
Fat-free mass (kg)	29.9 ± 4.1	33.6 ± 10.0	0.278	0.87	0.475	−0.3 to 1.2
Fat mass (%)	29.8 ± 17.4	26.1 ± 6.8	0.516	0.86	0.282	−0.5 to 1.0
PT Ext 60°/s (N·m)	87.9 ± 14.5	120.0 ± 24.7	0.001	>0.99	1.585	0.9 to 2.3
PT Flex 60°/S (N·m)	40.6 ± 7.2	57.8 ± 8.9	<0.001	>0.99	2.112	1.2 to 3.0
CR (Flex/Ext) (%)	46.4 ± 4.8	49.5 ± 9.4	0.332	0.87	0.424	−0.3 to 1.2

Data are expressed as mean ± standard deviation. CI: confidence interval. PT: peak torque; Ext: extensors; Flex: flexors; CR: conventional ratio.

## Data Availability

The data presented in this study are available on request from the corresponding author. The data are not publicly available due to privacy restrictions.

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
