# Peer review of "Comparison of Knee Muscular Strength Balance among Pre- and Post-Puberty Adolescent Swimmers: A Cross-Sectional Pilot Study"

_healthcare, 2023, doi:10.3390/healthcare11050744_

Round 1
Reviewer 1 Report
This study aimed to compare knee flexor strength, extensor strength, and strength balance ratio, called the conventional ratio (CR), between prepubertal and postpubertal swimmers of both sexes. According to the results of the study, author prepared the conclusion and stated that the flexor and extensor muscles were significantly stronger in postpubertal female and male swimmers than in prepubertal swimmers of the same sex, and the strengthening was balanced between agonist and antagonist muscles, once the CR did not change between the groups of the same sex. The founding in this study showed the mean values of CR were lower than those recommended in the literature and therefore, the authors recommend that coaches focus on strengthening the hamstring muscles of adolescent swimmers to maintain a CR higher than 60%, which is recommended to minimize the risk of knee injury.
Consistence among methodology, results, discussion and conclusion could be observed. However, the title of the study is to investigate the “influence of puberty” and so the results of this study may be collapsed by other factors such as age, body size, etc. Statistical analysis with co-variances may be considered in order to eliminate those factors on knee muscular strength.
The following items are suggested for further revision.
1. Line 158, please double check if the sample size is 22+34=56 or 55 (Line 86)
2. Line 170, please check if any typo for the girls’ sample size in Table 4
Author Response
Reviewer # 1
This study aimed to compare knee flexor strength, extensor strength, and strength balance ratio, called the conventional ratio (CR), between prepubertal and postpubertal swimmers of both sexes. According to the results of the study, author prepared the conclusion and stated that the flexor and extensor muscles were significantly stronger in postpubertal female and male swimmers than in prepubertal swimmers of the same sex, and the strengthening was balanced between agonist and antagonist muscles, once the CR did not change between the groups of the same sex. The founding in this study showed the mean values of CR were lower than those recommended in the literature and therefore, the authors recommend that coaches focus on strengthening the hamstring muscles of adolescent swimmers to maintain a CR higher than 60%, which is recommended to minimize the risk of knee injury.
Consistence among methodology, results, discussion and conclusion could be observed. However, the title of the study is to investigate the “influence of puberty” and so the results of this study may be collapsed by other factors such as age, body size, etc. Statistical analysis with co-variances may be considered in order to eliminate those factors on knee muscular strength.
Answer: Thank you for drawing our attention to this important point. As you observed consistence among methodology, results, discussion and conclusion, we chose to modify the title to better fit to the aim and statistical analysis of the study. Please let us know if this modification does not resolve your doubts in this matter.
The following items are suggested for further revision.
- Line 158, please double check if the sample size is 22+34=56 or 55 (Line 86)
Answer: Thank you for drawing our attention to this mistake. The male group was composed by 56 swimmers. The mistake has been corrected in the abstract and method sections.
- Line 170, please check if any typo for the girls’ sample size in Table 4
Answer: Thank you for drawing our attention to this point. The mistake has been corrected.
Reviewer 2 Report
This paper is very useful to introduce the relarionship between the changes in male and female changes after agonistic training as a possibile element that can ipotetically reduce the numbers or the entità of the accident during training.
The conclusions are now not supported by the data of the paper.
Author Response
Reviewer # 2
This paper is very useful to introduce the relarionship between the changes in male and female changes after agonistic training as a possibile element that can ipotetically reduce the numbers or the entità of the accident during training.
The conclusions are now not supported by the data of the paper.
Answer: Thank you for drawing our attention to this important point. The conclusion has been rewritten in order to clarify and meet with your expectation.
Reviewer 3 Report
It's a very correct methodological strategy with precise description.
I have only one remark. Conventional ratio (CR) was not precisely defined in the part of introduction.
Author Response
Reviewer # 3
It's a very correct methodological strategy with precise description.
Answer: Thank you about your positive evaluation.
I have only one remark. Conventional ratio (CR) was not precisely defined in the part of introduction.
Answer: The CR has been better defined in the introduction. Please let us know if these modifications do not resolve your doubts in this matter.